# Brazilian female researchers do not publish less despite an academic structure that deepens sex gap

Juliana Hipólito[1]*, Leila Teruko Shirai[2], Rosana Halinski[3], Aline Sartori Guidolin[4], Ranyse Barbosa Querino[5], Eliane Dias Quintela[6], Nivia da Silva Dias Pini[7], Carmen Sílvia Soares Pires[8], Eliana Maria Gouveia Fontes[8]

1 Instituto de Biologia, Universidade Federal da Bahia, Salvador, BA, Brasil, 2 Instituto de Biologia, Universidade Estadual de Campinas, Campinas, SP, Brasil, 3 Escola Politécnica, Pontifícia Universidade Católica do Rio Grande do Sul, Porto Alegre, RS, Brasil, 4 Escola Superior de Agricultura "Luiz de Queiroz", Universidade de São Paulo, Piracicaba, SP, Brazil, 5 Embrapa Cerrados, Planaltina, DF, Brazil, 6 Embrapa Arroz e Feijão, Santo Antônio de Goiás, GO, Brazil, 7 Embrapa Agroindústria Tropical, Fortaleza, CE, Brazil, 8 Embrapa Recursos Genéticos e Biotecnologia, Brasília, DF, Brazil

* juhipolito@gmail.com

**Data Availability Statement:** The datasets analyzed during the current study are available from the Capes portal at https://geocapes.capes.

## Abstract

In the 21st century, we still need to talk about gender inequality in science. Even with the sharp growth of studies on this theme over the last decades, we are still trying to convince our peers that diversity matters and, if embraced, makes better science. Part of this drawback can be related to the need for data to support effective proposals to change the academic scenario. In order to close some of those gaps, we here analyze 1) the profile of Brazilian researchers based on production, impact, and membership to the Brazilian Academy of Sciences, 2) participation in the Editorial boards of Brazilian journals dedicated to Entomology, and, 3) the academic scenario of Brazilian Entomology focusing on the sex of the first and last authors in peer-reviewed international publications related to Entomology. We aimed to provide a deeper look on the Brazilian Entomology scenario and to expand the amount of data availability to stimulate and foster a mind-change in the current academic structure. We performed scientometric searches and analysis using different platforms and found that the number and impact of the publications by female researchers, as observed by relative numbers, are not less than that of males. Despite that, female researchers are less represented at the Brazilian Academy of Sciences and editorial boards, reinforcing the lack of women recognition in science. Thus, we observe that some narratives related to the productivity gap can be misleading to a perpetuation of our internal and structural biases. We here expanded data from a previous paper where we scrutinized the Brazilian Entomology scenario and discussed the patches and systems that maintain gender gap in science.

## Introduction

The role of women scientists is prominent in several biological disciplines related to softer [1] or harder sciences [2]. However, women are still underrepresented in academic positions in

gov.br/geocapes/ and https://dadosabertos.capes.gov.br/dataset?organization=diretoria-de-avaliacao, and from Dimensions at https://www.dimensions.ai. At this last (Dimensions) some restrictions may apply considering terms and conditions. Data are however available from the authors upon reasonable request and permission of Dimensions (approved proposed project (DIM-024)) and can also be accessed through a personal or institutional registration at Dimensions portal. Thus, none of the used third party data had any special access privileges that others would not have.

**Funding:** We received financial support to realize this study by SEB the Sociedade Entomológica do Brasil as SEB is going to pay for open access if paper is accepted. SEB is a Brazilian organization which aims to support national researchers in Entomology. The funders had no role in study design, data collection and analysis, decision to publish, or preparation of the manuscript.

**Competing interests:** The authors have declared that no competing interests exist.

many fields [3, 4]. According to UNESCO Institute for Statistics, less than 30% of world researchers are female. The academic world is marked by a leaky pipeline, or scissor-shaped curve, as women are the majority in lower stages of academia (e.g. under graduation) but the minority on advanced career stages (e.g. professorship) [5]. Many recent publications are dealing with this theme, including women as well as other minorities (e.g. mothers, transgender, black) [5–9]. Like the general trend above for all sciences, despite women being the majority in lower degree stages in science, technology, engineering and math (STEM), they are the minority in higher degree stages (permanent positions and positions of prestige and power) [5]. In the 21st century, we still need to talk about gender inequality and we are still trying to convince our peers that diversity matters and, if embraced, makes better science [10–12]. Part of this drawback can be related to insufficient time to advance diversity and inclusion, lack of recognition for doing so [13], or personal or subjective difficulties in accepting differences or breaking stereotypes [14], but it may also be because we still need data to support effective proposals to change the academic scenario.

For example, it is well known that, despite the gender gap being everywhere, the amplitude and thus the impact can be different among disciplines and countries [15–17]. But even within historically male-dominated fields like Entomology [18], there is still insufficient data to understand how to address gender bias and how the available knowledge can contribute to proposals in other fields of science. Despite an increase in female first authorship at Entomology journals in the past years, their underrepresentation is still very evident and consistent [4]. But also, an increase of women as first authors does not reflect last authorships, which are usually related with senior researchers and lab leaders where we expect higher male dominance. Looking into both authorship positions could reveal a stronger pattern of gender underrepresentation.

Aside from the only two scientometric studies on Entomology, done in the USA [4, 19], the study of insects has been investigated in a single other country, Brazil, propelled by the two largest entomological societies in the world (the Entomological Society of America, ESA and Entomological Society of Brazil, SEB in its acronym in Portuguese). We have recently gathered information on how the Entomology scenario is structured in Brazil [5]. We did not include, however, the bigger picture of the gender gap in Entomology found in academic journals, patterns of co-authorship, publications impact, editorial boards, and the national Academy of Sciences, which are all aspects that, along with academic education and mentorship [5], internationally apply to what we call here as the academic structure.

One marked feature in modern science is that, despite much discussion on how to properly evaluate a scientist's quality, we still do it by measuring them with numbers [20]. Scientific publications are used as markers of merit [21] and prestige, increasing researchers' visibility in the academic community [22]. Highly productive and cited researchers tend to stand in front of new opportunities and receive more grants [23], which in turn generates a continuous loop of publications that feeds-back on implicit biases in peer review and citations [24, 25]. For example, the 'productivity puzzle', which refers to the supposed explanation for the underrepresentation of women because they are less productive, is still an open question [26]. Although some studies evidence a persistent lack of women and international diversity among top-publishing authors [27], with a lower number of articles than men in different areas of science [16, 17], they do not take into consideration, for example, that women have shorter career lengths and higher dropout rates [17].

Alternative metrics, or altmetrics, are non-traditional bibliometrics related to the scholar impact based on online research output, such as social media, instead of the usual citation counts [28]. Those metrics are advocated as a less biased evaluation of researchers who have career interruptions or primary caregiving responsibilities, as well as of early and mid-career

 

scientists [29]. An Altmetric analysis of eight major journals found no evidence of gender bias for seven of them [30]. This should be expanded to other journals and areas.

There are other proxies or metrics that are commonplace to evaluate a researcher's performance, e.g. being a manuscript reviewer or member of the editorial board of scientific journals. Editors-in-chief and editorial-board members, for example, are usually unpaid positions, but are rated as important since they can, among others, help researchers to reach longtime professional goals or have other benefits [31]. To become part of an editorial board, some reputation within a specialty or field is required [32], which is also prone to subjective biases as traditional metrics, but less transparently and thus less debatable–in fact, this may also apply to becoming a member of the Brazilian Academy of Sciences [11] and other scientific societies. This participation increases the experience and networking of a scientist, also allowing the gain of an insider's view of ways to improve a manuscript to facilitate or promote a fast track peer review process, i.e. gain in experience to the publication machinery, which could accelerate publication efforts with other journals and boost the marketability for research grants, hiring and promotions [31]. Editors' selection can reflect their hierarchic position in the academic rank and, in that manner, it can be used as a proxy for gender representation and leadership in academic spaces [32–34].

In sum, there are many aspects of the academic structure that are not related to science, but to the people doing it. Scientists, as humans, are biased and today, we are guided by an idea based on the male stereotype for what is a researcher [35, 36]. Thus, to provide further elements to motivate an increase of diversity in academic spaces, we analyzed the 1) the profile of Brazilian researchers based on production, impact, and membership in the Brazilian Academy of Sciences, 2) the participation of women in editorial boards of Brazilian journals dedicated to Entomology and, 3) the academic scenario of Brazilian Entomology focusing on the sex of first and last authors on peer-reviewed international publications related to Entomology. With this, we aimed to provide a deeper look at the Brazilian Entomology scenario and expand the amount of data to foster a change in the current academic structure.

## Material and methods

### Brazilian Entomologist profiles

To characterize the academic Brazilian scenario, we searched the Lattes curriculum database of the Brazilian National Council for Scientific and Technological Development (CNPq, from the name in Portuguese), through the iAraucaria platform (https://www.iaraucaria.pr.gov.br/intelectus/, accessed in May 2021) using the same keywords selected in a previous publication [5] that best represented Entomology for Brazilian studies. However, here we included Portuguese and English words, instead of only Portuguese as in our previous inquiry [5]:

"Archaeognatha OR Auchenorrhyncha OR Blattaria OR Blattodea OR Coleoptera OR Coleorrhyncha OR Collembola OR Dermaptera OR Dictyoptera OR Diplura OR Diptera OR Embioptera OR Ephemeroptera OR Grylloblattaria OR Grylloblattodea OR Hemiptera OR Heteroptera OR Homoptera OR Hymenoptera OR Isoptera OR Lepidoptera OR Mantodea OR Mantophasmatodea OR Mecoptera OR Megaloptera OR Neuroptera OR Odonata OR Orthoptera OR Phasmatodea OR Phasmida OR Phthiraptera OR Plecoptera OR Protura OR Psocodea OR Psocoptera OR Raphidioptera OR Siphonaptera OR Sternorryncha OR Strepsiptera OR Thysanoptera OR Trichoptera OR Zoraptera OR Zygentoma OR entom* OR inset* OR insect* OR bee OR abelha OR ant OR formiga OR Apis OR Drosophila OR fly OR mosca OR Spodoptera OR mosquito OR wasp OR vespa OR borboleta OR butterfly"

For this search we considered only researchers with a Masters (MSc) or Doctorate (PhD) degrees. The platform provides data on: 1) the highest degree (area, year, institution, and

level), 2) total journal articles, 3) total patents, 4) total completed adviserships (MSc and PhD), and 5) total academic productions (i.e., bibliographic and/or technical). We classified researchers as having female or male names in a two-steps process. We first used the *genderBR* package for R software [37] and second, we manually classified those names that the package was not able to automatically classify (mainly neutral names). For manually classified names, we tracked information on the researcher using the author's ORCID or webpage to double-check the author/publication. Although we recognize that a strict binary sex attribution is not the best, this simplifies the analysis of the representation of women in Entomology and can be compared to other studies, as we lack gender studies including other categories (e.g. trans, non-binary).

Additionally, we looked for Entomology researchers who are part of the Brazilian Academy of Sciences (ABC, from the name in Portuguese) to find out about the difference in prestige between the sex of these researchers on the national scene. The ABC has different categories for individual members as following: (i) full members–scientists living in Brazil for over ten years, with outstanding scientific expertise; (ii) corresponding members–scientists, recognized for scientific merit, living abroad for over ten years and who have provided relevant collaboration to the development of science in Brazil; (iii) affiliate members–young researchers of excellence, under 40 years old who are part of the ABC for a five-year period; and (iv) collaborating members–personalities who have provided relevant services to ABC or to national scientific development.

We searched ABC members in the main areas of Biology and Agronomy, and we individually checked their CNPq Lattes curriculum (http://buscatextual.cnpq.br/buscatextual/busca. do?metodo=apresentar, accessed on Sep 9th 2021) for their relation with Entomology. We considered the multiple fields of Entomology researchers, such as Agronomy, Biodiversity, Conservation, Ecology, Genetics, Public Health, and Zoology. Thus, we looked for works with insects in a broad sense (e.g. if the researcher works with insects independently from the main area: Zoology or Genetics for example). We excluded researchers that only contributed to entomological studies and thus do not have entomology as their field of expertise (i.e. researchers that were on insect trial commission or joined a research paper as a collaborator, not in a main position). On the selected Entomology researchers in the ABC, we assigned their sex (male or female) by the first name.

## Brazilian Entomology journals

To have a deeper understanding of Brazilian academic journals in Entomology, we selected every Brazilian journal with an exclusive content of Entomology: Sociobiology, Entomological Communications, "Revista Brasileira de Entomologia" [Brazilian Journal of Entomology], Entomobrasilis, and Entomology Beginners. We also included Neotropical Entomology on our list. Although not a Brazilian journal (based in the U.S.A.), it has great importance related to publications by Brazilian authors. Editors-in-chief are also Brazilian researchers (please see S1 Table for complementary details on those journals, including websites and national editors). On these, we analyzed the editorial board considering the sex, assigned by their first names, of all Editors-in-chief and Academic or Associate Editors that we could find on the journal's webpage.

On September 1st 2021, we performed a search in the Dimensions Website (https://app. dimensions.ai/exports) and explored all articles published specifically on these journals (except for Entomology Beginners that was not in this database, probably due to its recency). We only considered articles authored by scientists affiliated to an institution or organization based in Brazil, for at least one of the authors of the paper, regardless of the authorship position. On

those, we also classified male or female names using the two-steps process described above. We were not able to find 2.5% of publications or authors, mostly authors with abbreviated names that we did not find information in any other way, including our databases.

We analyzed both absolute and relative number of female and male names for comparisons of the number of papers and citations. We calculated the relative numbers by dividing the value of each sex for a certain year (for number of papers) or the value for the impact metric (for citations) by the total records of that sex, for example, relative(women) = women(year)/ women(all years). We compared absolute and relative numbers between sexes with the non-parametric Wilcoxon rank sum test, excluding missing values, because data was not normal (evaluated with the Lilliefors test and quantile-quantile QQ plots). We tested if there were sex differences related to the number of citations, number of co-authors on those papers, and alternative metrics to citations, such as the Relative Citation Ratio (RCR), Field Citation Ratio (FCR) and Altmetric. Those metrics measure the scientific influence of a publication, being complementary to traditional citation-based metrics as they can include (but are not limited to) peer reviews, citations on Wikipedia and in public policy documents, discussions on research blogs, mainstream media coverage, bookmarks on reference managers like Mendeley, and mentions on social networks such as Twitter. Specifically for comparisons between sexes year by year, we used $\chi^2$ test, with an expected ratio of 50%:50%. We performed all statistical analyses using the R software [38], except from a Word Cloud, done in the online tool Word-Clouds (https://www.wordclouds.com/). We used this tool to gain a visual idea of the most frequent words (nouns, adjectives, verbs, and numbers) in article titles (all titles in Portuguese were translated to English).

### Entomology papers by Brazilians in international journals

On September 15[th] 2021, we performed another search on the Dimensions Website using the same selected keywords as above. We limited our searches for articles in which the research organization and first or last authors affiliation was based in Brazil. We followed the same two-steps process to classify female and male names. We were not able to find, or were not certain, of 0.9% of publications or authors, again specially from authors with abbreviated names and that we did not find information in any other database. We found that the first publication dated from 1961, but since only 27 studies were published until 1997 with our search criteria, we decided to exclude publications prior to 1997. We performed the same analyses as above (Brazilian Entomology Journals).

## Results

### Brazilian Entomologist profiles

We found 7,958 researchers related to Entomology keywords searching at the iAraucaria database, with more researchers with a PhD as the highest degree than MSc (Table 1). Most of them were female researchers (52.7%) yet, with statistically significant (p<0.0001) lower numbers of published papers, patents, master and doctoral adviserships and other publications (books, chapters) (Table 1). Only technical publications or products had lower differences related to sex differences (p = 0.01). In this dataset, we observed that most male PhD researchers had obtained the title before females, that is, they had been researchers for a longer time (Fig 1). As most Entomology researchers and professors in Brazil must own a PhD degree to be hired today, that can be used as a proxy for which career stages the population of each sex are in the iAraucaria results.

Despite the higher number of female researchers in Entomology with MSc or PhD degrees (Table 1), we found a lower number of female researchers as members in the Brazilian

**Table 1. Summary of the iAraucaria and Wilcoxon rank sum test results (W and the p value, except for Msc or PhD degree).**

|  | Female | Male | W | p |
|---|---|---|---|---|
| PhD degree | 2,336 | 2,299 | - | - |
| MSc degree | 1,857 | 1,459 | - | - |
| Number of published papers | 57,453 | 93,304 | 9157556 | < 2.2e-16 |
| Patents | 634 | 894 | 8065584 | 0.0001 |
| MSc supervision | 7,410 | 12,174 | 8647372 | < 2.2e-16 |
| PhD supervision | 3,264 | 6,190 | 8564906 | < 2.2e-16 |
| Other publications | 206,180 | 268,169 | 8786174 | < 2.2e-16 |
| Technical publications or products | 94,509 | 100,734 | 8140102 | 0.016 |

Here we evidence the number of researchers with the highest degree as Doctorate (PhD) or Master's (MSc), and the number of published papers, patents, advisorships, and other publications related to researcher sex.

Academy of Sciences (ABC). From 218 members in all ABC members categories of Agrarian and Biological Sciences, we found 31 researchers which could be attributed to Entomology. Among these, we found a higher proportion of male researchers in all categories, for the full member category for example, there are ten full members and only two are female (Table 2, categories described in the Methods section). We did not find any collaborating member following our search criteria.

## Brazilian Entomology journals

We found 3,632 publications in Brazilian journals dedicated to Entomological topics. Most of paper titles included Hymenoptera (including many hymenopteran families) and Diptera (Fig 2A).

Authors contribution on those papers (related to the number of published papers by sex), evidenced an interesting pattern that we observed throughout our results. Analyzing absolute

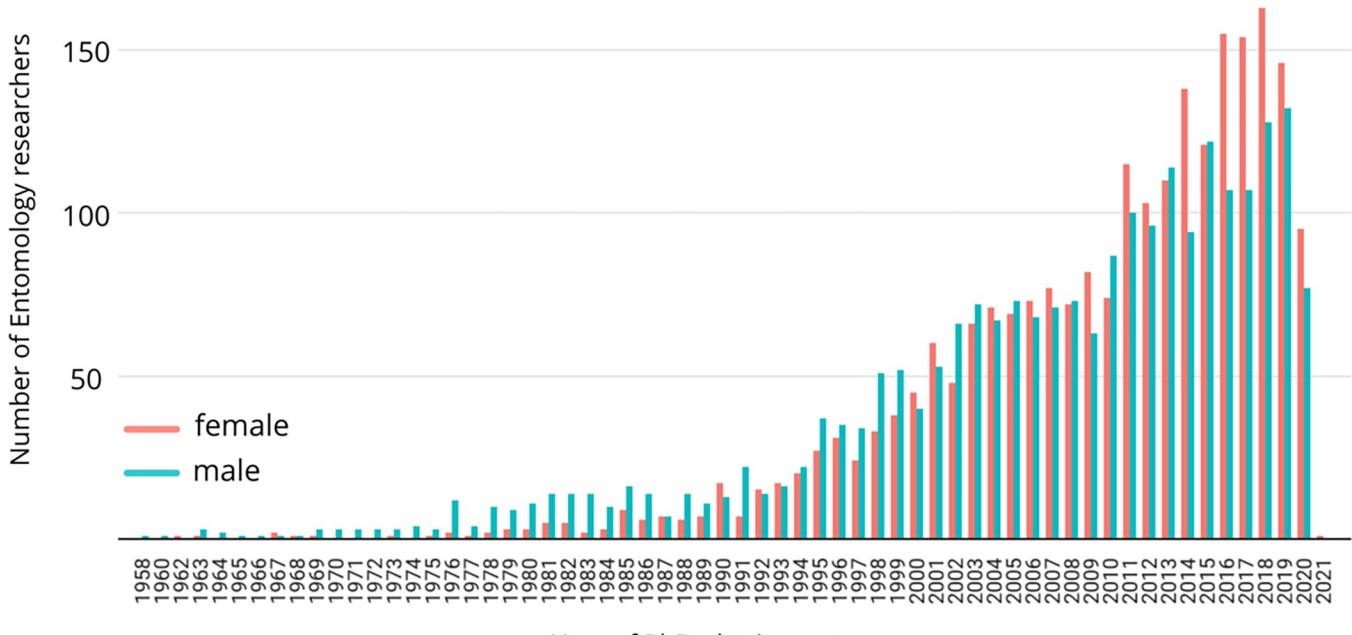

**Fig 1. Total number of female and male researchers per year of degree (PhD) obtainment in the iAraucaria platform.**

**Table 2. Male and female researchers as members of the Brazilian Academy of Sciences (ABC) considering different categories.** Proportions consider the total researchers in Entomology (30), numbers in parentheses are the absolute values.

| ABC category | Female researcher | Male researcher |
|---|---|---|
| Full Member | 6.4% (2) | 25.8% (8) |
| Corresponding Member | 0 | 16.1% (5) |
| Affiliate Member | 6.4% (2) | 41.9% (13) |

values, we observed a higher number of papers published when male researchers are in the first or last position (Fig 3, Table 3). However, this difference was not found in relative numbers in both first or last position (S2 Fig).

Impact metrics for Brazilian Entomology journals (number of citations, RCR, FCR and Altmetric) evidenced significant differences between sexes for absolute numbers only for FCR when male authors were in the first position and in all metrics despite Altimetric when they are in last position (Table 3). When we analyzed relative numbers for the statistically significant differences in these metrics, we found no such difference (S1 Fig, Table 3).

Brazilian Entomological editorial boards were mainly composed by male researchers (Fig 4). Only Neotropical Entomology has a female researcher as the current Editor in chief.

## Entomology papers by Brazilians in international journals

We found 14,586 articles in 1,295 journals from the keyword search in the Dimensions platform. We noticed that among all insect orders, studies on Diptera, Hymenoptera, Lepidoptera and Coleoptera were the most frequent (Fig 2B). We found the highest number of papers at the journal Neotropical Entomology (7.9%), followed by PLOS One, "Revista Brasileira de Entomologia," "Zoologia," Journal of Medical Entomology, Biota Neotropica, and Florida Entomologist, with all remaining comprising 12.2% of all articles.

In this larger dataset, we did not find statistically significant differences for the number of published papers between sexes for the first authorship position (Table 4). We did not find that impact metrics had a statistical difference between female and male first authors either (Table 4). In the single case of the FCR metric, male researchers showed a slight tendency of higher impact only when considering relative values (S1 Fig, Table 4).

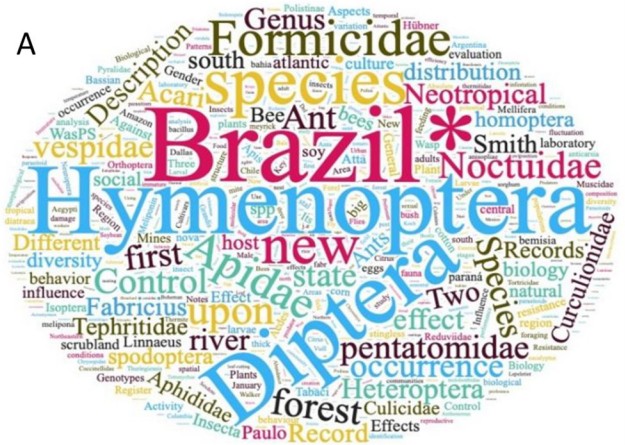
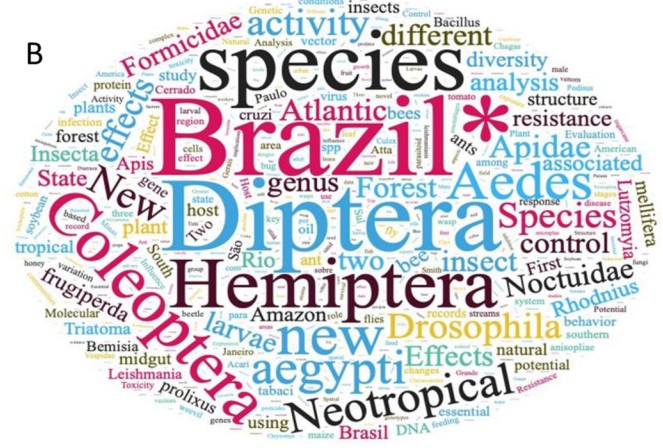

**Fig 2.** Word Cloud of the most frequent words on entomological articles found in the (A) Brazilian Entomology journals and (B) in international journals. Word size evidences the frequency of that word on paper titles. The * after Brazil denotes a suffix and thus includes Brazil and Brazilian.

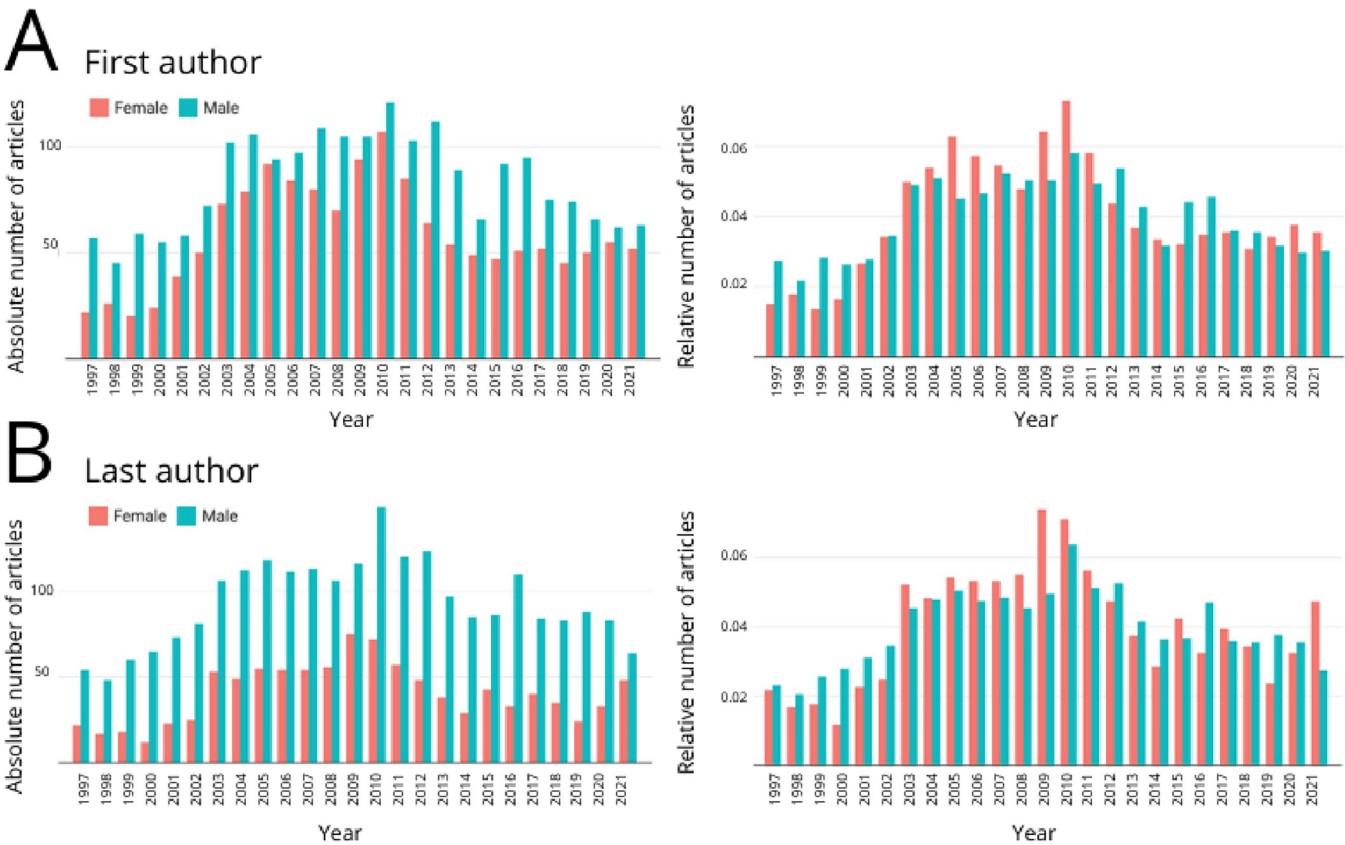

**Fig 3.** Number of articles authored by female and male researchers as (A) first authors and (B) last authors. Left panels evidence absolute numbers while right panels relative number of articles. We collected the data using Entomology keywords in the Dimensions database of Brazilian journals with an exclusive Entomology content, displaying the results from 1997 to 2021.

**Table 3. Traditional and alternative metrics of research papers considering absolute vs relative values between sexes in Brazilian Entomological journals.**

| 1st author | | | | |
|---|---|---|---|---|
| | W | p | W | p |
| | **absolute** | | **relative** | |
| Number of articles | 134 | **0.00055** | 319 | 0.9073 |
| Number of citations | 1189 | 0.2876 | 1364 | 0.9403 |
| RCR | 2097.5 | 0.2226 | 2652 | 0.2469 |
| FCR | 23422 | **0.002546** | 29110 | 0.3933 |
| Altmetric | 51 | 0.5501 | 63 | 0.8951 |
| Last author | | | | |
| Number of articles | 23 | **2.036e-08** | 314 | 0.9845 |
| Number of citations | 799 | **0.04306** | 1090 | 0.8056 |
| RCR | 1298.5 | **0.0003064** | 2327 | 0.1826 |
| FCR | 13132 | **1,20E-11** | 24755 | 0.1453 |
| Altmetric | 51 | 0.2258 | 89 | 0.3359 |

Metrics are related to impact (number of articles and number of citations) and alternative metrics to citations, such as the Relative Citation Ratio (RCR), Field Citation Ratio (FCR) and Altmetric. "W" on table refers to Willcox index sum test and "p" is related to probability values in statistical analyses, bold values highlight statistical significance between male and female authors.

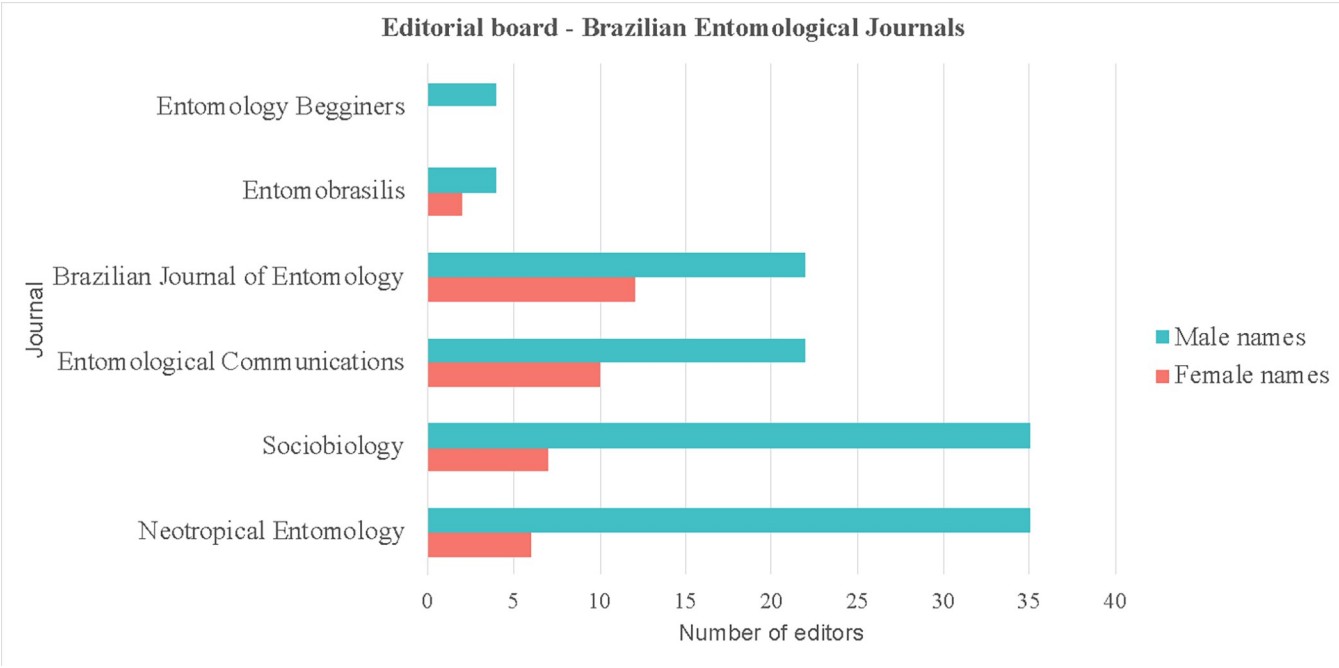

**Fig 4. Total number of researchers in the editorial board of the Brazilian journals dedicated to Entomology.**

The same was not the case when looking at researchers in the last position. While absolute numbers evidenced that there were statistically significant differences on the number of papers and on impact citation metrics, relative numbers showed that those differences do not occur (Table 4, Fig 5), except for the FCR metric that remained significantly different.

**Table 4. Traditional and alternative metrics of research papers considering absolute vs relative values between sexes in 1,295 International journals with Entomological keywords in the paper title.**

| 1st author | | | | |
|---|---|---|---|---|
| | W | P | W | p |
| | absolute | | relative | |
| Number of articles | 281.5 | 0.55 | 307 | 0.923 |
| Number of citations | 5877.5 | 0.42 | 6180 | 0.85 |
| RCR | 44125 | 0.89 | 47504 | 0.14 |
| FCR | 178900 | 0.33 | 197910 | **0.03** |
| Altmetric | 1221.5 | 0.24 | 1379 | 0.87 |
| Last author | | | | |
| Number of articles | 185.5 | **0.01** | 312 | 1.00 |
| Number of citations | 5342.5 | **0.01** | 6958 | 0.49 |
| RCR | 29284 | **5,36E-05** | 40431 | 0.73 |
| FCR | 115320 | **< 2.2e-16** | 182740 | **0.006** |
| Altmetric | 790.5 | **0.007** | 1139 | 0.93 |

Metrics are related to impact (number of articles and number of citations) and alternative metrics to citations, such as the Relative Citation Ratio (RCR), Field Citation Ratio (FCR) and Altmetric. "W" on table refers to Willcox index and "p" is related to probability values in statistical analyses, bold values highlight statistically significant differences between male and female authors.

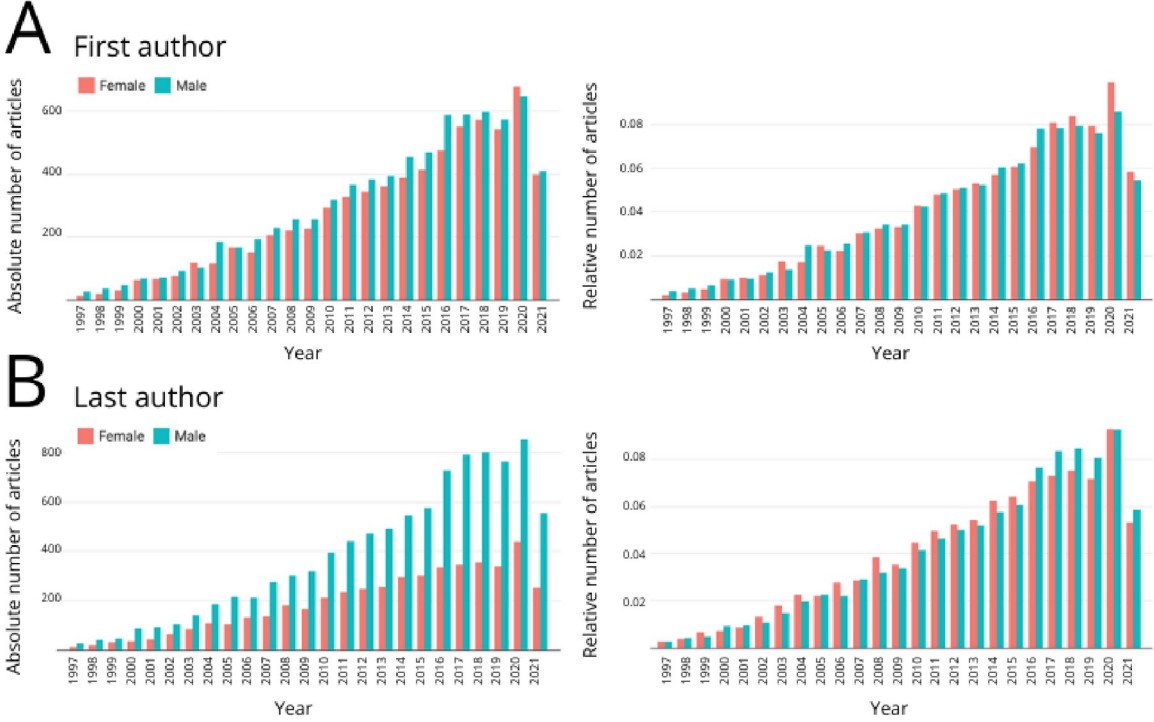

**Fig 5. Number of articles authored by female and male researchers as (A) first authors and (B) last authors.** Left panels evidence absolute numbers while right panels relative number of articles. We collected the data using Entomology keywords in the Dimensions database with an exclusive Entomology content, displaying the results from 1997 to 2021.

## Discussion

The number of studies related to the gender gap in academia has been sharply increasing over the last decades, with 3.6 million results in google scholar (searching, in Sep 4[th] 2021, for "gender gap" only, when there are many other ways of referring to the topic), with at least thousands of results in more structured searches [39, 40]. We find ourselves in a mid-point where there is much data available and, at the same time, we still lack any gender data for some regions or disciplines, or the data available is biased in its sampling or analysis. Furthermore, the gender gap has been investigated under different perspectives, such as historical ones, the leaky pipeline, mentorship and peer review bias, role stereotypes, productivity puzzle, strategies for inclusion and retention, etc. Ideally, having as many of these aspects systematically tackled for every discipline would contribute to a fuller understanding of, and more robust actions against, the gender disparity.

Earlier studies suggested that the highest number of academic outputs related to men could be explained by a lower productivity of women, the "productivity gender gap" reviewed in [40]. In Brazilian graduate courses dedicated to Entomology, for example, we found 86 female to 229 male professors [5]. It would only be expected that male entomologists produce more papers as they are the majority, and the majority in permanent positions (e.g. professors). Here, we analyzed a larger scope of academic productivity in Brazilian Entomology and, in absolute numbers, males did indeed dominate in the number of publications and impact, especially considering the last authorship position. However, looking at our results in relative terms, this dominance disappears.

It is rather surprising that the productivity gap is a claim often based in comparisons of absolute numbers (of papers, for example) [16, 17, 27], when it is widely documented that

there are more male researchers in science and in higher positions [26]. Research productivity should also consider time at each career stage and if there are enough women holding professional positions to perform those comparisons, otherwise it can be meaningless. Our results from the iAraucaria search showed a sex gap in publications, advisorships and patents, despite female researchers being the majority, because we included both MSc and PhD students and/or unemployed researchers or in temporary conditions (post-doctoral fellows). The latter stage was when we observed the leakiest point of the scissor-shaped curve [5]. Perhaps results of other studies, including our own (from the Dimensions database), might be influenced by uneven proportions of researchers at different career stages (or the age of degree as a proxy), showing the productivity gap not only because there are more male researchers, but also more male researchers hired in permanent positions and for longer periods [26]. Another relevant temporal factor would be to partition the data into samples of career lengths with equal time spans, as did [17], but as we did not have access to the end year of an individual career's, we cannot interpret this as a basis of the observed productivity gender gap.

We thus raise the possibility that some narratives drawn from productivity gap data can be quite misleading to, consciously or not, a perpetuation of our internal and structural biases. We did not observe that female authors publish less in relative terms, but we found that they are less recognized. We detected that the academic structure related to boards at scientific journals and prestigious communities as the Brazilian Academy of Sciences (ABC) evidenced a sex gap in a prominent manner. Thus, the diagnosis of the Brazilian Entomology scenario presented here revealed that female researchers do not publish less, but they appeared less in the academic scenario probably due to its structure. Not only can we observe this pattern at the ABC but at the 100,000 most influential scientists in the world [41], where only 0.25–0.30% are Brazilians [25, 41, 42] and, among these, only 11% of those were women [25]. Also, around 1% of scientists worldwide among the 100k most influential were assigned in the subfield of Entomology and, among them, seven are Brazilians, all male researchers [41, 42].

Awards, prizes and recognition in the scientific career trajectory play a major role in the gender gap stratification, enhancing the status of scientists who already have large reputations (the Matthew-Matilda Effect). It has also been demonstrated that, for example, while women are more equalitarian in collaborations, men are more likely to collaborate with other men [43]. Editors, authors, and reviewers are influential in shaping science, reflecting the high regard and trust in a community of colleagues [39]. But women are underrepresented in academic boards, specially at the editor-in chief position, even when women are equally productive as men, indicating a selection bias favoring men for editorial boards [44]. This same pattern was found not only for Brazilian Entomological journals or journals mainly composed by Brazilian researchers in Entomology (Neotropical Entomology) but also in the top ten journals in the Scimago Journal & Country Rank (S2 Table) [45]. Thus, this may probably reflect an international pattern [32–34, 44]. When there are less opportunities for recognition, it might seem that women have done less work than their male peers. This impression, which affects individual women as well as women worldwide, is not necessarily based on facts but it creates a real decrease in women's chance of advancement in academic and other positions [44].

The scientific community constantly considers as influential a scientist who publishes a high number of articles, with high number of citations [25]. However, the number of papers and of citations, often equalized with scientific impact, has been repeatedly shown not to reflect the quality of the scientific endeavor [12, 46]. For example, the H-index is highly sensitive to self-citations, a behavior typical of many male researchers [47], which then makes this metric reflect more self-confidence than scientific quality [25]. Yet, many gender gap studies use these metrics because they are easily obtainable and because we are evaluated by these

standards. We aimed at circumventing this issue by analyzing alternative metrics. Yet, we still have to investigate deeper those metrics. We observed, for example, that FCR had significant differences for absolute values in both first and last authorship positions, independent of what other metrics showed (traditional, RCR or Altmetric). However, this could be due to a statistical issue, because we had more papers with reported FCR values than the other metrics (S1 Fig). Altmetric on the other hand, did not show gender bias in any comparison (first and last author, absolute and relative numbers), so Altmetric can be related to online engagement and self-advertising, without necessarily reflecting gender-based patterns.

It is worth highlighting that although we had complementary data on our bibliometric results presented here and in [5], we did not entirely capture the career dynamics around teaching, administrative, industrial, or government-related research activities. Academia today has a controversy of teaching and evaluating students and post-docs to be individually successful, to be then hired in a position that requires the opposite skills, for example: leadership of research groups, administration of grants, projects, departments [48]. When students face this environment and, intentionally or not, choose for strategies to individually "win" and have "success" in the academic career, they may be driven to perpetuate current systemic structures without having time or will to question them. That goes for men and women who "made it", but the externality is that this structure seems to increase dropout rates from underrepresented groups who encounter more obstacles in their careers [49].

Even with affirmative actions in many universities for over a decade [16], the barriers to minorities remain and they will take long to dissipate if changes towards inclusion and diversity continue to be unrecognized, or continue to be a load done preferentially by minorities [13, 50], who already tend to overwork to compensate bias and achieve the same recognition as majorities [51–53]. Thus, the main question still is: where and how do we break this cycle?

One important place to start is the change in narrative, in our personal lives and in our community. For example, several authors point out that the productivity gap relates with factors such as family responsibility, indicating that marital status and having children are some of the aspects to be considered in this scale [47, 54, 55]. However, instead of placing such emphasis on women's burden and the structural blaming of mothers, the narrative could enhance alternative viewpoints such as that women with children of preschool age can have greater productivity compared to women with children of school age or without children, as they have a selective behavior and become skilled in making more appropriate time management [55]. Supporting diversity and parents in science is not an individual measure, we can have direct implications of those measures on our society. It is expected that children who are aware of gender differences since childhood and are raised accustomed with equally distributed opportunities and obligations between the parents and/or guardians grow to respect those values, which can lead to substantial changes in education and society, potentially impacting academic life and science as a whole.

We could also change discourses that explain women under representation based on the (lack of) choice of women to attend family responsibilities, or reinforcement of stereotypes based on merit or women inferiority. The scientific community could increase the focus to discussions on adopting strategies for hiring and retention of women and other minorities. Many studies have suggested excellent global and local actions to achieve gender equality, including how to diversify role models, how to make diversity statements work, how to reduce the motherhood penalty [12, 18, 39, 52, 56, 57]. Similarly, plentiful examples of propositions aiming to include other minorities (mothers, LGBTQIA+, Blacks, Hispanics, Asians, people with disability, from tropical countries, etc.) are already available [9, 50, 58, 59].

There are other measures that could be adopted as the inclusion of evaluation strategies beyond biased, simplistic, "unreliable, inaccurate, and damaging" [12] metrics like number of

publication and H-index, and to understand that taking time to compose a new cohort of professionals with more quality than quantity is an investment for better science and academic environments, even for individualistic researchers. These changes in evaluation can serve as a trigger for the propulsion of the common denominator in the women and other minorities movements: the "diversity and inclusion" moto. A central aspect of this moto is that diversity can only be built together with the minorities, and not for them [51, 58]; and diversity and inclusion need active energy to gain momentum, which must be proportional to the forces against it.

Main areas (like Exact sciences) [15, 17] or disciplines (as Ecology and Evolution) [13]; or Entomology, as shown here and in [4, 19] that are strongly male-biased might suffer more if we, as a community, do not admit that hegemony of any kind does not make better science. Still, one may point out that inequality shall not be entirely solved by a simple establishment of proportions [60]. Just like in evolutionary processes, there is a need to adapt to remain viable, and multivariate diversity and flexibility are natural strategies to avoid reaching stressful and costly dead-ends. The roadblocks to talented, but unprivileged people will remain if we do not acknowledge the major bias against them. Once the community is ready to change, very simple actions can have great impact, like deliberately choosing a female keynote speaker (with similar quality as the usually chosen male researcher) in a conference, or preferentially considering non-white junior women researchers for editorial boards, or extending curriculum evaluation or other eligibility criteria (such as the extension on time after obtaining a PhD) if a person became a mother. Other, better structured, venues of action also help, like hiring minorities to handle minority issues, or at least recognizing (in tenure tracks, grants, prizes, etc.) researchers who already work towards integrating diversity in their institutions.

We have previously argued [5] that top institutions should embrace the gender equity discourse for the new generations they inspire. Researchers in these institutions frequently serve as role models for academics and lay people, but one aspect that might particularly influence Brazil is that many senior university professors have not been hired in the same system that junior researchers are being hired now. In Brazil, it was not until the end of the dictatorship (in 1985) that the current hiring system ("editais de concursos públicos") began to be applied, before being through invitations. This would be, for example, like senior engineers deciding on the workings of the public transport but who themselves never take a bus or a subway. This can have a strong impact on the support that senior researchers see fit for the diversity and inclusion movements. There can be two venues about this, one is related to the justification of an 'innocent' ignorance as the lack of knowledge and other to a politically and socially strategy to maintain a system of privileges [9, 61]. While we allow it to be maintained for whatever reason, we will maintain the systematic biased exclusion of some groups and favoring others.

Another aspect that is relevant to the Brazilian Entomology scenario is that many senior researchers are in Agronomy Sciences, which is somewhat reflected in the most frequent orders found in paper titles (Fig 1). The megadiverse insect orders Diptera, Hymenoptera, Lepidoptera and Coleoptera are studied due to its immense diversity in Brazil, but more importantly because they act as agricultural pests or in biological control. But relevant to the discussion here is that Agronomy has been shown to be more male-biased than other Biological Sciences [5], and that can even affect how the conservation of biodiversity is led in the megadiverse country of Brazil.

## Conclusions

Expanding data and providing other effective proposals in order to have more diversity and inclusion is necessary [62]. Pathways that incentivize union of powers rather than competitive

exclusion are urgent. We demonstrated here that female researchers are not less productive than male researchers as relative numbers evidenced similar number of publications and citations. Yet, we still have a system that is based on recognizing male achievement over female achievement. The diversity in STEM goes beyond numbers and promotion. Its existence is necessary since when more young people and community members can see themselves in the scientist speaking, the message resonates with increased impact [29].

## Supporting information

**S1 Fig. Impact metrics for Entomology journals.** Impact metrics: number of citations, RCR, FCR and Altmetric, considering both male and female researchers occupying first and last positions on papers. Absolute and Relative values are demonstrated side by side.
(PDF)

**S2 Fig. Male to female ratios by year with absolute (Fig 1) and relative (Fig 2) values.** Fig 1. Male to the female ratio by year with absolute data considering one as an equal ratio value. Values higher than one evidence more male researchers and lower evidence more female researchers as first or last authors. Fig 2. Male to the female ratio by year with relative data considering one as an equal ratio value. Values higher than one evidence more male researchers and lower evidence more female researchers as first or last authors.
(DOCX)

**S1 Table. General information on Brazilian Entomological journals.** Brazilian Entomology journals considering journal metrics and editorial board members.
(XLSX)

**S2 Table. Top ten journals in the Scimago Journal & Country Rank within the subject area of Insect Science.** Top ten journals in Scimago Journal & Contry Rank considering journals Title, rank position and general information (Issn, H index, Country, region, publisher. . .), as well as the editorial proportion within those journals, related to Editors-in-chief or editorial board.
(XLSX)

**S1 File.**
(DOCX)

## Acknowledgments

We would like to thank the Dimensions platform for granting access to their data, Project: DIM-024, and Instituto Stella for helpful information on the iAraucaria database.

## Author Contributions

**Conceptualization:** Aline Sartori Guidolin, Ranyse Barbosa Querino, Eliane Dias Quintela, Nivia da Silva Dias Pini, Carmen Sílvia Soares Pires, Eliana Maria Gouveia Fontes.

**Data curation:** Juliana Hipólito, Leila Teruko Shirai, Rosana Halinski.

**Formal analysis:** Juliana Hipólito, Leila Teruko Shirai, Rosana Halinski.

**Investigation:** Juliana Hipólito.

**Methodology:** Juliana Hipólito, Leila Teruko Shirai, Rosana Halinski.

**Supervision:** Juliana Hipólito, Eliana Maria Gouveia Fontes.

**Writing – original draft:** Juliana Hipólito, Leila Teruko Shirai.

**Writing – review & editing:** Juliana Hipólito, Leila Teruko Shirai, Rosana Halinski, Aline Sartori Guidolin, Ranyse Barbosa Querino, Eliane Dias Quintela, Nivia da Silva Dias Pini, Carmen Sílvia Soares Pires, Eliana Maria Gouveia Fontes.

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
