## [Decision Letter · Decision Letter 0]

28 Mar 2022

PONE-D-21-36534Female researchers do not publish less despite an academic structure that deepens sex gapPLOS ONE

Dear Dr. Hipólito,

Thank you for submitting your manuscript to PLOS ONE. After careful consideration, we feel that it has merit but does not fully meet PLOS ONE’s publication criteria as it currently stands. Therefore, we invite you to submit a revised version of the manuscript that addresses the points raised during the review process.

We look forward to receiving your revised manuscript.

Kind regards,

António M. Lopes, PhD

Academic Editor

PLOS ONE

Journal Requirements:

“NO - The funders had no role in study design, data collection and analysis, decision to publish, or preparation of the manuscript.”

Reviewers' comments:

Reviewer's Responses to Questions

**Comments to the Author**

1. Is the manuscript technically sound, and do the data support the conclusions?

Reviewer #1: Yes

Reviewer #2: Yes

2. Has the statistical analysis been performed appropriately and rigorously? 

Reviewer #1: Yes

Reviewer #2: Yes

3. Have the authors made all data underlying the findings in their manuscript fully available?

Reviewer #1: No

Reviewer #2: Yes

4. Is the manuscript presented in an intelligible fashion and written in standard English?

Reviewer #1: Yes

Reviewer #2: Yes

5. Review Comments to the Author

Reviewer #1: The paper presents very interesting work about the gap between men and women in academic positions and is worthy of publication. Additionally, in terms of writing, it is fluid and clear.

However, some changes can be made to improve the manuscript. First, when the researchers in ABC and in Entomology in general are presented (Table 1 and 2), it would be interesting to establish a plot that would relate them. The same is valid for when the first/last authors per year, dividing men and women (Figure 5), are presented. Additionally, in Figure 5, it would be interesting to clarify if the relative number of articles is defined by the ratio between articles written by women and women+men, for each year or between the number of articles written by women that year and the total number of articles written by women.

Finally, a smaller and concise conclusion section would enrich the text.

Some additional comments can be found in the appended pdf file.

Reviewer #2: Paper

Female researchers do not publish less despite an academic structure that deepens sex gap

Comments

Interesting and current topic applied to a specific area of investigation. It can inspire other researchers from other areas to carry out similar studies to verify if the conclusions obtained are similar in other areas of knowledge.

Very careful description of Materials and Methods.

Extensive list of bibliographic references and recent publications.

Very good results discussion.

Minor fixes:

1. Check the entire text “Brazilian Academy of Science” or ”Sciences”.

2. Any explanation for the peak of publications to be between the years 2003 to 2012 and remain more stable from there?

3. Table 2 - absolute values sum 30 and not 31 as indicated in the text.

4. Conclusions are missing. some sentences from the discussion should be included in a new “Conclusions” section.

6. PLOS authors have the option to publish the peer review history of their article (what does this mean?). If published, this will include your full peer review and any attached files.

Reviewer #1: No

Reviewer #2: No

---

## [Author Response · Author response to Decision Letter 0]

26 May 2022

Dear Editor,

We want to thank you and the two anonymous reviewers for the valuable and constructive

feedback on the submitted manuscript. We have addressed most of reviewer’s comments. Some of them we did not, yet we justified below. Please see our detailed responses (in blue) to each comment (in black). In addition to addressing the suggestions by the reviewers.

We hope that the revised manuscript fulfils the expectations for publication in PlosOne.

Best regards,

Juliana Hipólito and colleagues

Comments from Reviewer 01: 

Interesting and current topic applied to a specific area of investigation. It can inspire other

researchers from other areas to carry out similar studies to verify if the conclusions obtained are

similar in other areas of knowledge.

Very careful description of Materials and Methods.

Extensive list of bibliographic references and recent publications.

Very good results discussion.

RESPONSE: Thank you for your valuable and constructive feedback. We are very glad about all positive comments. 

Minor fixes:

1. Check the entire text “Brazilian Academy of Science” or ”Sciences”. Done 

2. Any explanation for the peak of publications to be between the years 2003 to 2012 and remain more stable from there?

For any scientific field there is an increase of publications, as we showed for example in Fig 5 (for all journals), which makes this question pertinent because it does not follow this general trend. The comment refers to Fig 3, related to the Brazilian journals. The increase in 2003 is partly related to the general trend, also partly because the Revista Brasileira de Entomologia appears in the dataset in 2002 - another "jump" happens in 2008 with the addition of data from EntomoBrasilis in the database. The number of articles indeed lessened by 2012, but it does not relate to removal of journals in the dataset. The most relevant journals, Revista Brasileira de Entomologia and Neotropical Entomology, both decrease their number of papers by 2012. We are not completely sure for what generate those patterns yet, seems to have some explanations that are beyond our objectives, although super interesting to be analyzed in future studies. 

3. Table 2 - absolute values sum 30 and not 31 as indicated in the text. Corrected 

4. Conclusions are missing. some sentences from the discussion should be included in a new

“Conclusions” section. We included a conclusions section as suggested. Thank you very much. 

Comments from Reviewer 02: 

Please consider explaining why women are referred to as a minority, since they represent the majority of the world's population. 

RESPONSE: We included more explanation on that on the first paragraph. 

 A trendline of the average could be interesting

RESPONSE: We appreciate reviewer comment and we made the legend clear. However we do not consider that we can add a trendline of average as in each year we have the total number of researchers instead of a population data in each year. 

A plot that describes the relation between researchers in ABC and researchers in Entomology would be interesting

Response: We appreciate reviewer comments, yet we already made this comparison in a previous paper [1] where we included this relation between researchers in different academic stages. As this was the focus of our previous paper, and at this present paper, this could be incomplete, we would instead not include a plot that described this relation here. 

1. Hipólito J, Shirai LT, Halinski R, Guidolin AS, da Silva Dias Pini N, Soares Pires CS, et al. The Gender Gap in Brazilian Entomology: an Analysis of the Academic Scenario. Neotrop Entomol. 2021 [cited 12 Nov 2021]. doi:10.1007/s13744-021-00918-7

It would be interesting to add a plot of first author/last author, or similar, to establish a relation

RESPONSE: We really appreciate reviewer suggestion. We included this plot as a supplementary file 

Figure 5. Please standardize. Please clarify if relative(women)=women/(women+men) or relative(women)=women(year)/women(all years)

RESPONSE: Absolutely, this is very relevant. As the same calculation was used for all relative values we report on the paper, we added the equation to the Material and Methods section, where it was already written but perhaps not so clearly. If the editor and/or reviewer think(s) it is relevant to add to the figure legend, we add during proofs to both Figure 3 and 5

---

## [Decision Letter · Decision Letter 1]

15 Jul 2022

PONE-D-21-36534R1Female researchers do not publish less despite an academic structure that deepens sex gapPLOS ONE

Dear Dr. Hipólito,

Thank you for submitting your manuscript to PLOS ONE. After careful consideration, we feel that it has merit but does not fully meet PLOS ONE’s publication criteria as it currently stands. Therefore, we invite you to submit a revised version of the manuscript that addresses the points raised during the review process.

We look forward to receiving your revised manuscript.

Kind regards,

António M. Lopes, PhD

Academic Editor

PLOS ONE

Journal Requirements:

Reviewers' comments:

Reviewer's Responses to Questions

**Comments to the Author**

1. If the authors have adequately addressed your comments raised in a previous round of review and you feel that this manuscript is now acceptable for publication, you may indicate that here to bypass the “Comments to the Author” section, enter your conflict of interest statement in the “Confidential to Editor” section, and submit your "Accept" recommendation.

Reviewer #1: All comments have been addressed

Reviewer #3: (No Response)

Reviewer #4: All comments have been addressed

2. Is the manuscript technically sound, and do the data support the conclusions?

Reviewer #1: Yes

Reviewer #3: Yes

Reviewer #4: Yes

3. Has the statistical analysis been performed appropriately and rigorously? 

Reviewer #1: Yes

Reviewer #3: Yes

Reviewer #4: Yes

4. Have the authors made all data underlying the findings in their manuscript fully available?

Reviewer #1: Yes

Reviewer #3: No

Reviewer #4: Yes

5. Is the manuscript presented in an intelligible fashion and written in standard English?

Reviewer #1: Yes

Reviewer #3: Yes

Reviewer #4: Yes

6. Review Comments to the Author

Reviewer #1: The authors made the necessary changes to the paper and it is now suitable for publication. As a final remark, the reviewer suggests using gender instead of sex in the title, since the social term "gender" is more suitable than the biological "sex", to the reviewer's best knowledge.

Reviewer #3: The paper “Female researchers do not publish less despite an academic structure that deepens sex gap” describes gender inequality in Brazilian science in terms of the presentation of women in key / authoritative positions despite the fact of equal productivity from a bibliometrics viewpoint. The results are based on a solid ground formed from three sources: (a) the structure of the Brazilian Academy of Science; (b) the composition of editorial boards of the Brazilian journals; and (c) the positions of males/females in the bylines of papers published by Brazilian researchers. The sample is limited by Entomology subject area; bibliometric analyses were carried out using the Dimensions database. Applied approaches are thoroughly detailed, as well as the samples used.

The paper is believed to contribute to our knowledge on gender distribution and productivity not only because it concerns poorly studied region and one more subject area, but also due to substantive discussion of what should be made to increase inclusivity, diversity, and equality. The authors proposed several steps including raise of awareness of the problem, introducing some normalized values considering motherhood, the age of children, other family responsibilities, etc. to increase reasonable recognition of male/female contribution in science. I think the revised version of the manuscript is worth publishing after addressing several minor issues.

Comments.

1. Title. Now the title lacks both mentioning entomology subject area and Brazilian science. I think they should be included in the title to make it more precise.

2. Pp. 14-15, Brazilian journals description. It will be useful to add some more information on Brazilian entomological journals, like their ISSNs, websites, indexing systems, the language of the articles, etc., to make readers’ perceptions more informed. As I understand, some Brazilian journals may be international ones, include international members, and thus may represent not only Brazilian, but also international models of gender distribution in entomology. Thus, a more detailed explanation of the compound of editorial boards in Brazilian journals is needed. The other interesting point is to compare male/female distribution in Brazilian journals and, e.g., top-10 international entomological journals. For instance, in 10.3103/S0147688218040123 it was found that women were underrepresented in a set of analyzed national journals on Earth Sciences while international ones followed other models. Is there Brazilian specific in this aspect?

3. P. 22: “if we, as a community, do not admit that hegemony of any kind does not make better science.” – However, sometimes the causes of inequality are reasonable and should not be solved by a simple establishment of proportions. See, e.g., 10.1016/S0140-6736(03)12528-7.

4. P. 9: “Embrapa Arroz e Feijão, Santo Antônio de Goiás, GO, Brazil.” – remove dot character in the end of the line.

5. P. 10: “degree stages in STEM” – please, explain the abbreviation on first mention.

6. P. 14: The reference to the Dimensions Website is indicated twice in this page. I believe the second mention can be removed.

Reviewer #4: The topic is interesting and the new version is understandable and easy to read. All the changes requested by the reviewers were made.

7. PLOS authors have the option to publish the peer review history of their article (what does this mean?). If published, this will include your full peer review and any attached files.

Reviewer #1: No

Reviewer #3: No

Reviewer #4: No

---

## [Author Response · Author response to Decision Letter 1]

25 Jul 2022

Dear Editor, we really appreciate reviewers comments although a few we do not agree. Below we comment 

Review Comments to the Author

Reviewer #1: The authors made the necessary changes to the paper and it is now suitable for publication. As a final remark, the reviewer suggests using gender instead of sex in the title, since the social term "gender" is more suitable than the biological "sex", to the reviewer's best knowledge.

Answer: We appreciate reviewer comments, but we, unfortunately, did not access gender in our study as gender is socially constructed and is not binary. As we just identified names instead of a person’s response, we do not see as the most appropriate term the use of term “gender.” 

Reviewer #3: The paper “Female researchers do not publish less despite an academic structure that deepens sex gap” describes gender inequality in Brazilian science in terms of the presentation of women in key / authoritative positions despite the fact of equal productivity from a bibliometrics viewpoint. The results are based on a solid ground formed from three sources: (a) the structure of the Brazilian Academy of Science; (b) the composition of editorial boards of the Brazilian journals; and (c) the positions of males/females in the bylines of papers published by Brazilian researchers. The sample is limited by Entomology subject area; bibliometric analyses were carried out using the Dimensions database. Applied approaches are thoroughly detailed, as well as the samples used.

The paper is believed to contribute to our knowledge on gender distribution and productivity not only because it concerns poorly studied region and one more subject area, but also due to substantive discussion of what should be made to increase inclusivity, diversity, and equality. The authors proposed several steps including raise of awareness of the problem, introducing some normalized values considering motherhood, the age of children, other family responsibilities, etc. to increase reasonable recognition of male/female contribution in science. I think the revised version of the manuscript is worth publishing after addressing several minor issues.

Answer: We really appreciate reviewer comments

Comments.

1. Title. Now the title lacks both mentioning entomology subject area and Brazilian science. I think they should be included in the title to make it more precise.

Answer: We included Brazilian at the title and entomology at the keywords. As the title is already a bit larger we’d rather not to include entomology at the title.

2. Pp. 14-15, Brazilian journals description. It will be useful to add some more information on Brazilian entomological journals, like their ISSNs, websites, indexing systems, the language of the articles, etc., to make readers’ perceptions more informed. 

Answer: We added this information on a supplementary table. Supplementary material 3.

3. As I understand, some Brazilian journals may be international ones, include international members, and thus may represent not only Brazilian, but also international models of gender distribution in entomology. Thus, a more detailed explanation of the compound of editorial boards in Brazilian journals is needed.

Answer: We added more information on this point at the material and methods. We agree with reviewer point of view. 

4. The other interesting point is to compare male/female distribution in Brazilian journals and, e.g., top-10 international entomological journals. For instance, in 10.3103/S0147688218040123 it was found that women were underrepresented in a set of analyzed national journals on Earth Sciences while international ones followed other models. Is there Brazilian specific in this aspect?

Answer: We added another supplementary material with the top ten international journals form SciMago (supplementary material 4) and expand this discussion on the discussion section. 

3. P. 22: “if we, as a community, do not admit that hegemony of any kind does not make better science.” – However, sometimes the causes of inequality are reasonable and should not be solved by a simple establishment of proportions. See, e.g., 10.1016/S0140-6736(03)12528-7.

Answer: I agree with reviewer comment and really appreciate suggestion. I added a note about it: “if we, as a community, do not admit that hegemony of any kind does not make better science. Still one may point that the causes of inequality are reasonable and should not be completely solved by a simple establishment of proportions [59].”

4. P. 9: “Embrapa Arroz e Feijão, Santo Antônio de Goiás, GO, Brazil.” – remove dot character in the end of the line.

Answer: Excluded

5. P. 10: “degree stages in STEM” – please, explain the abbreviation on first mention.

Answer: We included the explanation

6. P. 14: The reference to the Dimensions Website is indicated twice in this page. I believe the second mention can be removed.

Answer: We excluded the second as suggested

Reviewer #4: The topic is interesting and the new version is understandable and easy to read. All the changes requested by the reviewers were made.

Answer: We appreciate reviewer comments

---

## [Editor Report · Decision Letter 2]

8 Aug 2022

Brazilian female researchers do not publish less despite an academic structure that deepens sex gap

PONE-D-21-36534R2

Dear Dr. Hipólito,

We’re pleased to inform you that your manuscript has been judged scientifically suitable for publication and will be formally accepted for publication once it meets all outstanding technical requirements.

Kind regards,

António M. Lopes, PhD

Academic Editor

PLOS ONE
---

## [Editor Report · Acceptance letter]

19 Aug 2022

PONE-D-21-36534R2 

Brazilian female researchers do not publish less despite an academic structure that deepens sex gap 

Dear Dr. Hipólito:

I'm pleased to inform you that your manuscript has been deemed suitable for publication in PLOS ONE. Congratulations! Your manuscript is now with our production department. 

Kind regards, 

on behalf of

Dr. António M. Lopes 

Academic Editor

PLOS ONE